## Research Article

sensory processing; self-regulation; executive function; autism; attention deficit-hyperactivity disorder; preschool children

**Corresponding author:**
Sabide Duygu Uygun;
Email: sduygun@ankara.edu.tr

# Exploring self-regulation deficits in sensory over-responsivity disorder: A preschool comparative analysis

Sabide Duygu Uygun [ID], Esma Kara [ID], Rahime Duygu Temeltürk, Esra Yürümez, Merve Cikili Uytun and Didem Behice Öztop

Department of Child and Adolescent Psychiatry, Ankara University School of Medicine, Ankara, Turkey

## Abstract

Sensory Over-Responsivity Disorder (SORD) is characterized by extreme sensitivity to everyday sensory input, which can interfere with children's emotional, behavioral and social development. Despite growing interest, limited research has explored its developmental effects in the absence of other psychiatric diagnoses. This study investigated self-regulation and related clinical features in preschool children with SORD who did not meet diagnostic criteria for autism spectrum disorder, attention-deficit/hyperactivity disorder (ADHD), or obsessive-compulsive disorder. The sample included 15 children with SORD and 15 typically developing controls, matched by age and gender. Diagnoses were made using the Preschool Age Psychiatric Assessment, and comorbidities were excluded using Diagnostic Classification of Mental Health and Developmental Disorders of Infancy and Early Childhood: Revised Edition criteria. Self-regulation was assessed through the Head-Toes-Knees-Shoulders-Revised task. While no significant differences were found in autistic traits, repetitive behaviors or executive functioning, children with SORD demonstrated significantly poorer self-regulation ($p < .001$). Poorer self-regulation was strongly associated with greater SORD severity, elevated ADHD symptoms, lower social interaction and increased emotional and sensory reactivity. These findings suggest that self-regulation difficulties are a core feature of SORD, even in the absence of comorbid psychiatric disorders. Early identification and interventions targeting self-regulation may help improve long-term outcomes for children affected by SORD.

## Impact statement

Sensory Over-Responsivity Disorder (SORD) is a condition where children have intense negative reactions to ordinary sensory inputs—such as sounds, textures or smells. These responses can significantly interfere with daily activities at home, in school and in social settings. Despite its impact, SORD is not formally recognized in major diagnostic systems, and little is known about its effect on essential developmental skills like self-regulation.

This study is among the first to investigate self-regulation in preschool children diagnosed with SORD but without comorbid psychiatric conditions. We found that these children exhibited significantly greater difficulties in regulating their emotions and behaviors compared to typically developing peers. Importantly, these deficits were independent of other symptoms such as hyperactivity or repetitive behaviors. The severity of sensory sensitivity was strongly associated with problems in emotional control and social interaction.

These findings are important because early self-regulation difficulties are known predictors of long-term academic and mental health issues. Demonstrating that SORD alone can lead to these challenges highlights the need for early identification and support. Our results suggest that self-regulation difficulties may be a core feature of SORD and should be considered in its clinical profile.

This research draws attention to an often-overlooked group and underscores the importance of developing targeted, early interventions to support children with sensory over-responsivity and improve their long-term developmental outcomes.

## Introduction

Sensory Over-Responsivity Disorder (SORD) is characterized by an exaggerated sensitivity to sensory stimuli, which can significantly impair a child's daily functioning. According to the Diagnostic Classification of Mental Health and Developmental Disorders of Infancy and Early Childhood: Revised Edition (DC:0–5 classification) (Zero to Three, 2016), children with SORD exhibit intense and aversive reactions to one or more types of ordinary sensory input across

multiple environments, including home, preschool and playground settings. These heightened responses often lead to avoidance behaviors and can negatively affect the child's social relationships, participation in age-appropriate activities and overall developmental trajectory. Importantly, for a diagnosis of SORD to be valid, these symptoms must not be better explained by another psychiatric disorder, such as autism spectrum disorder (ASD) or attention-deficit/hyperactivity disorder (ADHD) (Zero to Three, 2016). The absence of a dedicated diagnostic category for SORD in major classification systems, such as the Diagnostic and Statistical Manual of Mental Disorders (DSM) and the International Classification of Diseases, poses additional challenges to clinical recognition and diagnostic clarity (American Psychiatric Association, 2013; World Health Organization, 2019).

Over time, sensory over-responsivity (SOR) has primarily been examined in clinical populations, where it frequently co-occurs with neurodevelopmental disorders such as ASD and ADHD, rather than being recognized as a distinct diagnostic entity (Istvan et al., 2020; Keating et al., 2022; Niedźwiecka et al., 2020). It has been associated with core behavioral symptoms and social difficulties commonly observed in these conditions, suggesting that SOR may function as a nonspecific marker of broader psychopathology (Cheung and Siu, 2009). Supporting this view, prior research has indicated that children exhibiting elevated levels of ADHD symptomatology demonstrate significantly higher SOR scores compared to those with low or absent ADHD symptoms (Ben-Sasson et al., 2017; Lane et al., 2010; Reynolds et al., 2010), as well as those with elevated autistic traits or formal ASD diagnoses (Ben-Sasson et al., 2019). More recently, SOR has also been identified in internalizing disorders, including anxiety and obsessive-compulsive disorder (OCD), further expanding its clinical relevance beyond neurodevelopmental profiles (Cervin, 2023). These findings have led to the proposition that SOR may represent an early-emerging, nonspecific phenotype that transcends categorical diagnoses and is relevant across multiple psychiatric conditions (van den Boogert et al., 2022). Notably, SOR has also been observed in individuals without any formal psychiatric diagnoses, underscoring substantial gaps in our understanding of its developmental origins and diagnostic boundaries (Little et al., 2017; Yuan et al., 2022).

In the context of SORD, sensory-seeking behaviors – such as repetitive or compulsive actions – may not reflect a desire to obtain additional sensory input, but instead serve as compensatory strategies to regulate heightened arousal levels (Keating et al., 2022; Tal et al., 2023). These behaviors may represent attempts to restore sensory homeostasis in the presence of overwhelming environmental input. This view is supported by meta-analytic findings, suggesting that sensory-seeking behaviors among individuals with sensory modulation difficulties, including SOR, follow nonlinear developmental trajectories and vary depending on factors such as cognitive level, age and self-awareness (Ben-Sasson et al., 2019). For instance, seeking behaviors tend to peak in early childhood (ages 6–9 years) and decrease thereafter, indicating their possible role as early self-regulatory mechanisms before the development of more mature coping strategies. Therefore, in children with SORD, sensory-seeking behaviors may be expected as an early-emerging self-regulatory strategy, particularly in the absence of more adaptive emotional regulation mechanisms. These behavioral patterns reflect underlying sensory processing thresholds and are closely tied to individual self-regulation capacities (Dunn, 2007; Little et al., 2017). As a core executive function, self-regulation enables children to manage emotional and behavioral responses to both internal and external stimuli (Ros and Graziano, 2020). Yet, despite growing attention, the relationship between sensory processing and self-

regulation remains underexplored (DeGangi et al., 2000; Lai et al., 2019; Previtali et al., 2023). Sensory processing, governed by an individual's neurological threshold, is essential for effective behavioral regulation and adaptation to environmental demands (Dunn, 2001; 2007). Dunn's model emphasizes that sensory-seeking and avoiding behaviors emerge as adaptive or maladaptive attempts to manage internal arousal. Given the clinical significance of these associations, investigating self-regulatory functioning in children with SORD is essential for developing targeted interventions aimed at improving adaptive functioning and overall quality of life (Dunn, 2007; Yuan et al., 2022).

On this basis, the objective of the present study is to compare self-regulation skills, emotional and behavioral problems, autistic traits, repetitive behaviors and executive functions between preschool children with SORD without psychiatric comorbidities (such as ASD, ADHD or OCD) and healthy controls. More specifically, the study aims to examine the relationship between self-regulation and the severity of SOR, while accounting for other clinical variables. The following hypotheses are proposed: (i) Preschool children with SORD (without psychiatric comorbidities) will exhibit significantly more emotional and behavioral problems compared to healthy controls. (ii) Children with SORD will demonstrate significantly poorer self-regulation skills and executive functioning (EF) than healthy controls. (iii) Children with SORD are expected to display significantly more intense sensory-seeking behaviors, including repetitive and compulsive behaviors, which may function as compensatory strategies to manage heightened arousal. (iv) The severity of SOR will be significantly associated with self-regulation deficits and repetitive behaviors, independent of autistic traits and emotional or behavioral problems.

## Methods

The research protocol was approved by the Human Research Ethics Committee (Approval number: İ05–383-2024), and the cross-sectional study was conducted in accordance with the Declaration of Helsinki between July and October 2024 at an academic Infant Mental Health Unit. A total of 24 preschool children were initially diagnosed with SORD; however, 9 were excluded due to the presence of psychiatric comorbidities based on the DC:0–5 classification (Zero to Three, 2016), resulting in a final SORD group of 15 children.

The control group also consisted of 15 typically developing children matched for age and sex. These children were recruited from the university-affiliated preschool center through informational flyers. The flyers explained the purpose of the study and invited families to participate in a free, comprehensive psychiatric and developmental evaluation. Contact details of the principal investigators were provided, and families who voluntarily expressed interest were subsequently invited to the clinic for clinical interviews and assessments. Only children without any psychiatric diagnosis were included in the control group.

### Procedure

Patients aged between 3 and 6 years, who applied to our outpatient clinic, were evaluated, and written and verbal consent was obtained from the parents who agreed to participate in the study. Following this, a child psychiatrist interviewed the parents and children. During the psychiatric evaluation, the intelligence levels of all children were assessed to be within clinically normal limits. All children underwent the Childhood Autism Rating Scale (CARS) to

rule out a diagnosis of ASD, and none met the criteria for the disorder. SORD diagnosis and severity were assessed with the Preschool Age Psychiatric Assessment (PAPA). All parents completed sociodemographic data forms, the Strengths and Difficulties Questionnaire (SDQ), the Social Responsiveness Scale (SRS), the Childhood Executive Functioning Inventory (CHEXI) and the Repetitive Behavior Scale-Revised (RBS-R). All children were evaluated using a brief self-regulation assessment, the Head-Toes-Knees-Shoulders-Revised tasks (HTKS-R).

### Measures

#### Sociodemographic data form
The form was created by the researchers to gather relevant sociodemographic information about the sample. It includes questions about the child's age, sex and preschool educational status. Additionally, it collects data on the parents' ages, educational levels and family income.

#### Preschool Age Psychiatric Assessment (PAPA)
The PAPA is a structured psychiatric interview designed for parents of children aged 2–6 years (Egger et al., 1999). This assessment uses the DSM, Fifth Edition, and DC:0–5 classification to identify psychiatric symptoms in early childhood. While the PAPA comprises 26 modules that cover various diagnostic areas, our focus was on the Regulation module. This module is designed to evaluate multiple aspects of sensory and emotional regulation among preschool-aged children. It investigates sensory processing abnormalities and includes a significant component (subscale) focused on SORD, which we utilized to confirm the diagnosis and assess its severity. A pivotal study validating the reliability and effectiveness of the PAPA in a Turkish context was conducted (Oztop et al., 2024). Each kappa value for all diagnoses is >0.6.

#### Childhood Autism Rating Scale (CARS)
The CARS is widely used for diagnosing autism and differentiating children with autistic disorder from those with other developmental disorders (Schopler et al., 1980). The scale is completed based on information gathered from family interviews and direct observation of the child. It consists of 15 items and employs a half-point rating system that ranges from 1 (*within normal limits*) to 4 (*severely abnormal*), evaluated through both observation and interview data. The items on the scale encompass various areas, including interpersonal relationships, imitation, emotional responses, use of the body, use of objects, adaptation to change, visual responses, listening responses, sensory modalities (taste, smell and touch), fear and nervousness, verbal communication, nonverbal communication, activity level, consistency of intellectual responses and general impressions. A score of ≥30 indicates the presence of autism. The Cronbach's alpha coefficient for its Turkish adaptation was .95 (Gassaloğlu et al., 2016).

#### Strengths and Difficulties Questionnaire (SDQ)
The SDQ, which includes both parent and teacher forms, is used to screen for emotional and behavioral problems in children and adolescents. The questionnaire consists of 25 items that assess both positive and negative behavioral attributes, divided into five subscales: (1) conduct problems, (2) hyperactivity and inattention, (3) emotional symptoms, (4) peer relationship problems and (5) prosocial behaviors (Goodman, 2001). It employs a three-point Likert scale for responses, where items are rated as follows: 0 indicates "*not true*," 1 signifies "*somewhat true*" and 2 represents "*certainly true.*" Each subscale is evaluated individually, and the sum of the first four subscales provides a total difficulties score. The Turkish version of the SDQ has been shown to be valid and reliable, with a Cronbach's alpha coefficient of .73 (Güvenir et al., 2008).

#### Social Responsiveness Scale (SRS)
The SRS is utilized to assess social behavior and autistic traits. The scale consists of 65 items that are organized into five subscales: (1) Social Awareness, (2) Social Cognition, (3) Social Communication, (4) Social Motivation and (5) Autistic Mannerisms (Constantino et al., 2003). Each item is rated on a four-point Likert scale, where 0 indicates "*not true*," 1 "*sometimes true*," 2 "*often true*" and 3 "*almost always true.*" The total score on the SRS ranges from 0 to 195, with higher scores representing greater severity of social impairment. Additionally, a validity and reliability study of the Turkish version of the scale was conducted for children aged 3–18 years, yielding a Cronbach's alpha coefficient of .86 (Ünal et al., 2009).

#### Childhood Executive Function Inventory (CHEXI)
The CHEXI is designed to assess children's EF skills as reported by their parents and teachers (Thorell and Nyberg, 2008). The inventory originally included 26 items divided into two sub-dimensions: "working memory" and "inhibitory control." Responses are measured on a five-point Likert scale, with ratings ranging from 1 ("*Definitely not true*") to 5 ("*Definitely true*"). Higher scores on the CHEXI indicate greater difficulties with EF skills, while lower scores suggest better EF. In its Turkish adaptation, the number of items was reduced to 24 while preserving the two-factor structure (Çiftçi et al., 2020; Hamamcı et al., 2021). The Turkish version of the inventory demonstrates high internal consistency, with Cronbach's alpha coefficients of .95 for the working memory subscale and .91 for the inhibitory control subscale (Çiftçi et al., 2020).

#### Repetitive Behavior Scale-Revised (PBS-R)
The RBS-R is a clinical rating scale designed to assess repetitive behaviors and their severity (Bodfish et al., 2000). The scale comprises six subscales: (1) Stereotyped Behavior (6 items), (2) Self-Injurious Behavior (8 items), (3) Compulsive Behavior (8 items), (4) Routine Behavior (6 items), (5) Sameness Behavior (11 items) and (6) Restricted Behavior (4 items), totaling 43 items. Items on the scale are rated using a four-point scale: 0 (*Behavior absent*), 1 (*Mild*), 2 (*Moderate*) and 3 (*Severe*). Higher scores indicate a greater severity of repetitive behaviors in children. The Turkish validity and reliability study of the scale was conducted for children aged 3–23 years (Ökcün Akçamuş et al., 2019). Reliability analyses revealed that the Cronbach's alpha values for all subscales are at least .73, while the overall score achieves a Cronbach's alpha coefficient of .94.

#### Head-Toes-Knees-Shoulders-Revised Tasks (HTKS-R)
The HTKS-R was developed as a measurement tool to assess the behavioral self-regulation skills of children aged 3–7 years (Ponitz et al., 2009) and has recently been revised (McClelland et al., 2021). This assessment tool consists of three sections, each containing 10 tasks, for a total of 30 tasks. The HTKS-R evaluates children's abilities to utilize attention, working memory and inhibitory control skills, while also promoting appropriate behavior in social interactions. The tasks require minimal training and no specialized materials, relying on the interaction between the practitioner and the child. In this context, children are expected to respond behaviorally to four different verbal commands, with their responses carefully observed and documented. Turkish validity and reliability studies of the HTKS

and HTKS-R were conducted (Ertürk Kara et al., 2024), yielding a Cronbach's alpha coefficient of .96 (Sezgin and Demiriz, 2015).

### Statistical analysis

The sample size was determined using G*Power 3.1.9.2 software, aiming for 80% power, which indicated a total of 30 participants (15 per group) aged 3–6 years (Faul et al., 2007; Golshan et al., 2019). Statistical analyses were performed using SPSS software version 23.0. Variables were assessed for normality using both visual methods (histograms and probability plots) and analytical methods (Shapiro–Wilk test). Descriptive analyses were presented as mean (standard deviation [SD]) and median with the first and third quartiles for continuous variables. Categorical variables were expressed as frequency (*n*) and percentage (%).

Correlation coefficients and statistical significance were calculated using the Spearman test to investigate the relationships between variables. For comparisons of groups, an independent sample *t*-test or Mann–Whitney *U*-test was used for independent samples, while comparisons between categorical variables were performed using the Chi-square test.

A univariate analysis of covariance was conducted to assess the effect of the independent variable (group: patient/control) on the dependent variable (HTKS-R total score), while controlling for potential confounding effects of the covariates: SDQ hyperactivity and inattention subscale score, RBS-R total score and SRS total score. This method aimed to examine group differences more precisely by controlling for specific covariate effects and isolating the impact of the group on the HTKS-R total score, thereby providing more reliable and valid results. A *p*-value < .05 was considered statistically significant.

### Results

The average age of the sample was 54.8 months (SD = 9.29), with each group consisting of nine males and six females. Table 1 presents the sociodemographic characteristics of preschool children with SORD and healthy controls. The groups were well-matched in terms of age, sex and socioeconomic status, minimizing potential confounding variables. Psychiatric assessments confirmed that, aside from SORD, no other psychiatric disorders were present. Children with SORD were identified using the SORD subscale of PAPA and matched by age and sex with peers without SORD. Clinical characteristics were then compared between the groups, revealing that, except for symptoms of attention deficit/hyperactivity (which were greater in the SORD group, *p* = .009), other emotional and behavioral difficulties were similar in both groups (see Table 2). Executive functions, such as working memory and inhibition, along with autistic traits and repetitive behaviors (including compulsions), were similar between the groups (all *p* > .05). Although the CARS was used to rule out ASD in children, a comparison of subscale features between the groups revealed that, as expected, responses related to taste, smell and touch were more pronounced in the SORD group (*p* = .01). Surprisingly, their ability to relate to people was also worse (*p* = .02), while other features were similar (all other *p* > .05). Regarding self-regulation skills, the SORD group performed worse on the HTKS-R, both overall and in all sections (see Table 2, all *p* < .001).

Clinical characteristics related to self-regulation and SOR severity in preschool children are presented in Table 3. SOR severity is represented by the total score on the SORD subscale of the PAPA,

**Table 1.** Sociodemographic characteristics of preschool children with sensory over-responsivity disorder and healthy controls

|  | SORDs | | Controls | | |
| --- | --- | --- | --- | --- | --- |
|  | Mean/% | SD/*n* | Mean/% | SD/*n* | *p* |
| Age/months | 55.2 | 9.67 | 54.4 | 9.2 | .740 |
| Sex |  |  |  |  | 1.00 |
| Female | 40% | 6 | 40% | 6 |  |
| Male | 60% | 9 | 60% | 9 |  |
| Preschool education |  |  |  |  | .409 |
| Yes | 66.7% | 10 | 80% | 12 |  |
| No | 33.3% | 5 | 20% | 3 |  |
| Maternal age/yrs. | 36 | 3.98 | 35.4 | 4.63 | .531 |
| Maternal education |  |  |  |  | .523 |
| None | 0% | 0 | 0% | 0 |  |
| Primary | 6.7% | 1 | 0% | 0 |  |
| Middle | 0% | 0 | 6.7% | 1 |  |
| High school | 13.3% | 2 | 20% | 3 |  |
| Graduate | 80% | 12 | 73.3% | 11 |  |
| Paternal age/yrs. | 38.67 | 4.65 | 38.6 | 5.82 | .819 |
| Paternal education |  |  |  |  | .592 |
| None | 0% | 0 | 6.7% | 1 |  |
| Primary | 0% | 0 | 0% | 0 |  |
| Middle | 0% | 0 | 0% | 0 |  |
| High school | 26.7% | 4 | 26.7% | 4 |  |
| Graduate | 73.3% | 11 | 66.7% | 10 |  |
| Family income (TL) | 101.933 | 84.353 | 71.333 | 52.388 | .371 |

Abbreviations: SD, standard deviation; SORDs, preschoolers with sensory over-responsivity disorder; yrs., years.

while self-regulation is represented by the total score on the HTKS-R. SOR severity and self-regulation were strongly correlated with each other (*r* = −.82, *p* < .001) and moderately associated with attention deficit/hyperactivity symptoms (*r* = .46, *p* = .009; *r* = −.40, *p* = .02, respectively), social interactions (*r* = .42, *p* = .02; *r* = −.52, *p* = .003, respectively) and sensory responses (*r* = .53, *p* = .003; *r* = −.42, *p* = .01, respectively). Additionally, self-regulation was linked to emotional responses (*r* = −.43, *p* = .015).

Covariance analysis, controlling for potential confounding factors such as attention deficit/hyperactivity symptoms, autistic traits and repetitive behaviors, confirmed the significance of the difference in self-regulation skills between the groups (see Table 4, *p* < .001). This underscores the robustness of the observed effect, independent of these other factors.

### Discussion

Our study reveals significant deficits in self-regulation skills of preschool children with SORD, independent of attention deficit/hyperactivity symptoms, repetitive behaviors and autistic traits, as confirmed by covariance analysis. We demonstrated a strong correlation between SOR severity and self-regulation, underscoring the

**Table 2.** Comparison of clinical characteristics and self-regulation between preschool children with sensory over-responsivity disorder and healthy controls

| | SORDs | | Controls | | |
|---|---|---|---|---|---|
| | Mean | SD | Mean | SD | *p* |
| SDQ | | | | | |
| Total | 10.4 | 5.58 | 8.4 | 4.26 | .479 |
| Emotional | 1.07 | 1.22 | 1.47 | 1.51 | .436 |
| Conduct | 2.27 | 2.05 | 2.13 | 1.85 | .949 |
| Hyper./Inatt. | 4.6 | 2.56 | 2.4 | 1.96 | .009* |
| Peer | 2.47 | 1.36 | 2.4 | 2.06 | .831 |
| Prosocial | 6.6 | 1.5 | 6.2 | 1.52 | .446 |
| CHEXI | | | | | |
| Working memory | 25.8 | 9.68 | 22.67 | 7.93 | .405 |
| Inhibitory control | 30 | 8.59 | 25.47 | 5.77 | .077 |
| SRS | | | | | |
| Total | 43.4 | 21.4 | 48.47 | 14.34 | .418 |
| Social awareness | 7.8 | 3.86 | 9.07 | 2.6 | .427 |
| Social cognition | 9.93 | 5.75 | 9.6 | 4.81 | .868 |
| Social comm. | 12.53 | 7.69 | 14.27 | 4.2 | .430 |
| Social motivation | 7.4 | 3.5 | 9 | 3.36 | .179 |
| Autistic mannerisms | 5.73 | 3.63 | 6.53 | 4.19 | .691 |
| RBS-R | | | | | |
| Total | 16 | 10.03 | 19.53 | 7.61 | .191 |
| Stereotyped behavior | 1.27 | 1.62 | .80 | 1.42 | .300 |
| Self-injurious behavior | .53 | .74 | .20 | .56 | .121 |
| Compulsive behavior | 3.8 | 3.75 | 7 | 4.71 | .063 |
| Routine behavior | 4.13 | 1.77 | 5.2 | 2.11 | .125 |
| Sameness behavior | 4.67 | 3.81 | 5.87 | 2.97 | .225 |
| Restricted behavior | 1.6 | 2.75 | .47 | .83 | .299 |
| CARS | | | | | |
| Total | 15.8 | 1.1 | 15.1 | .47 | .023* |
| Relating to people | 1.13 | .23 | .93 | .26 | .023* |
| Imitation | 1 | 0 | 1 | 0 | 1 |
| Emotional response | 1.17 | .24 | 1.03 | .13 | .073 |
| Body use | 1.03 | .13 | 1.07 | .18 | .550 |
| Object use | 1.03 | .13 | 1 | 0 | .317 |
| Adaptation to change | 1.1 | .21 | 1.03 | .13 | .291 |
| Visual response | 1 | 0 | 1 | 0 | 1 |
| Listening response | 1 | 0 | 1 | 0 | 1 |
| Taste, smell, touch Rsp. | 1.2 | .32 | 1 | 0 | .016* |
| Fear and nervousness | 1.13 | .3 | 1.03 | .13 | .276 |
| Verbal Comm. | 1 | 0 | 1 | 0 | 1 |
| Nonverbal Comm. | 1 | 0 | 1 | 0 | 1 |
| Activity level | 1 | 0 | 1 | 0 | 1 |
| Intellectual response | 1 | 0 | 1 | 0 | 1 |

(Continued)

**Table 2.** (*Continued*)

| | SORDs | | Controls | | |
|---|---|---|---|---|---|
| | Mean | SD | Mean | SD | *p* |
| General impressions | 1 | 0 | 1 | 0 | 1 |
| HTKS-R | | | | | |
| Total | 26.47 | 13.11 | 50.27 | 4.56 | < .001*** |
| Part 1 | 13 | 3.61 | 17.93 | 1.58 | < .001*** |
| Part 2 | 9.07 | 5.91 | 16.93 | 2.28 | < .001*** |
| Part 3 | 5.5 | 5.42 | 15.4 | 3.18 | < .001*** |

*Note:* Statistical significance levels: *p < .05, **p < .01, ***p < .001.
Abbreviations: SD, standard deviation; SORDs, preschoolers with sensory over-responsivity disorder; SDQ, Strengths and Difficulties Questionnaire; Hyper./Inatt., hyperactivity/inattention subscale; CHEXI, Childhood Executive Functioning Inventory; SRS, Social Responsiveness Scale; RBS-R, Repetitive Behavior Scale-Revised; CARS, Childhood Autism Rating Scale; Comm., communication; Rsp., response; HTKS-R, Head-Toes-Knees-Shoulders-Revised tasks.

distinct impact of SOR on self-regulation. Furthermore, our findings suggest that, even in the absence of other psychiatric comorbidities, SOR severity and self-regulation are moderately associated with attention deficit/hyperactivity symptoms and social interactions, with a notable link between self-regulation and emotional responses. These insights deepen our understanding of the unique challenges faced by preschool children with SORD and highlight the importance of targeted interventions to support their self-regulation development.

The DC:0–5 classification system is the first framework to define SORD as a disorder; however, its criteria are described quite non-specifically, highlighting the need for clearer definitions and specific clinical descriptions to accurately characterize this condition (Zero to Three, 2016). Unlike the DC:0–5 system, the Fifth Edition of the DSM (DSM-5) does not formally recognize sensory processing disorders. However, within the DSM-5 framework, SOR is noted as a potential symptom in the context of the repetitive and restricted behavior criteria for ASD. It is crucial to highlight that SOR should not be restricted solely to symptoms of ASD; rather, it must be regarded as an independent clinical entity that can manifest in individuals with various psychiatric disorders, as well as in those without any psychiatric conditions, thus warranting further investigation. Consequently, the clinical implications of SORD and its association with behavioral self-regulation remain significantly under-researched (Brout et al., 2018; Lai et al., 2019). This study represents the first investigation of SORD, specifically excluding psychiatric comorbidities and focusing on behavioral self-regulation in the preschool period. In SORD, weak self-regulation emerges independently of attention deficit/hyperactivity symptoms, repetitive behaviors and autistic traits; therefore, symptoms of impaired self-regulation should be included within the diagnostic criteria for SORD. Furthermore, examples of weak behavioral self-regulation observed in daily life (such as difficulty transitioning between activities, throwing temper tantrums when faced with unexpected changes or struggling to wait for their turn) should be articulated as part of the diagnostic criteria, similar to those identified in the HTKS-R tasks (Lai et al., 2019; Previtali et al., 2023).

The neurobiological basis of our findings regarding SOR and self-regulation may be better understood in light of research, indicating that SOR is associated with functional brain connectivity

**Table 3.** Investigating clinical characteristics related to self-regulation and sensory over-responsivity in preschool children

| | | PAPA-SORD/ Total | HTKS-R/ Total |
|---|---|---|---|
| HTKS-R/Total | r | −.820 | 1.000 |
| | p | <.001*** | |
| SDQ/Total | r | .146 | −.068 |
| | p | .442 | .720 |
| SDQ/Hyper./Inatt. | r | .467 | −.405 |
| | p | .009** | .026* |
| SRS/Total | r | −.111 | .085 |
| | p | .558 | .657 |
| RBS-R/Total | r | −.210 | .158 |
| | p | .265 | .403 |
| CARS/Total | r | .451 | −.509 |
| | p | .012* | .004* |
| CARS/Relating to people | r | .424 | −.520 |
| | p | .020* | .003** |
| CARS/Emotional response | r | .320 | −.439 |
| | p | .085 | .015* |
| CARS/Taste-smell-touch response | r | .530 | −.429 |
| | p | .003** | .018* |

*Note:* Statistical significance levels: *p < .05, **p < .01, *p < .001.
Abbreviations: PAPA-SORD, Preschool Age Psychiatric Assessment, Sensory Over-Responsivity Disorder subscale; HTKS-R, Head-Toes-Knees-Shoulders-Revised tasks; SDQ, Strengths and Difficulties Questionnaire; Hyper./Inatt., hyperactivity/inattention subscale; SRS, Social Responsiveness Scale; RBS-R, Repetitive Behavior Scale-Revised; CARS, Childhood Autism Rating Scale.

deficits in both individuals with and without ASD (Schwarzlose et al., 2023; Yuan et al., 2022). These studies have identified alterations in functional connectivity within sensorimotor networks (Green et al., 2018; Schwarzlose et al., 2023). Such disruptions can lead to difficulties in filtering and integrating sensory information. As a result, individuals may experience challenges with selective attention and inhibition of external stimuli, which could diminish their ability to inhibit excessive responses to incoming sensory inputs. Moreover, this disruption in inhibition control may correlate with difficulties in both behavioral and emotional regulation, emphasizing the intricate relationship between sensory processing challenges and poorer self-regulatory skills in children with SORD.

Specifically, a deeper understanding of the physiological basis of these behaviors may be found in research exploring autonomic nervous system functioning. Studies have shown that children with ASD, a group commonly affected by SOR, exhibit atypical autonomic functioning – such as diminished parasympathetic tone and exaggerated sympathetic reactivity – during both rest and sensory processing tasks (Bal et al., 2010; Neuhaus et al., 2014; Schaaf et al., 2010; Schaaf et al., 2015). These alterations in autonomic regulation may impair emotional and behavioral control in response to environmental demands. Although our study excluded children with ASD, it is plausible that similar subclinical autonomic dysregulation mechanisms contribute to the emotional reactivity and self-regulation deficits observed in SORD. These findings align with Porges' polyvagal theory, which posits that vagal tone influences one's ability to modulate behavior under stress (Porges et al., 2013). Future research incorporating physiological measures, such as respiratory sinus arrhythmia (RSA) or skin conductance, may further elucidate the biological underpinnings of sensory dysregulation and self-regulatory capacity in SORD.

From a conceptual standpoint, our findings reinforce the view that self-regulation difficulties in children with SORD are rooted in both behavioral and neurophysiological processes. As outlined by Dunn's Four Quadrant Model of Sensory Processing, self-regulation strategies are closely tied to one's neurological threshold and response style, producing characteristic sensory profiles, such as Sensory Sensitivity or Sensory Avoiding (Dunn, 1997). This framework provides a lens through which to interpret the behavioral manifestations observed in our SORD sample – namely, that children may adopt avoidance, resistance or dysregulated emotional reactions in response to overstimulation due to their low neurological threshold.

Moreover, this sensory-based perspective aligns conceptually with Eysenck's arousal theory of personality, which suggests that individuals actively seek or avoid stimulation to maintain optimal arousal levels (Sato, 2005). In both models, self-regulation is understood as a dynamic strategy to achieve internal balance in the face of environmental demands. This convergence reinforces the interpretation of our results: self-regulatory impairments in children with SORD may reflect deeper disruptions in their capacity to modulate arousal in response to ordinary sensory input.

Physiological research lends additional support to this hypothesis. Impaired sensory gating – such as deficits in P50 event-related potentials – has been observed in children with sensory processing challenges, limiting their ability to filter irrelevant stimuli and contributing to difficulties in attention, emotional modulation and behavioral inhibition (Davies et al., 2009; Davies and Gavin, 2007). These findings highlight the biological plausibility of our observed behavioral associations and suggest that interventions

**Table 4.** Investigating the effects of sensory over-responsivity, as well as autistic traits, repetitive behaviors and hyperactivity/inattention symptoms on self-regulation in preschool children

| | Sum of squares | Degrees of freedom | Mean square | F | p |
|---|---|---|---|---|---|
| SDQ/ Hyper.-Inatt. | 14.635 | 1 | 14.635 | .137 | .714 |
| SRS/Total | 26.312 | 1 | 26.312 | .246 | .624 |
| RBS-R/Total | 14.569 | 1 | 14.569 | .136 | .715 |
| Group (SORDs vs. Controls) | 3,049.473 | 1 | 3,049.473 | 28.561 | < .001*** |
| Error | 2,669.268 | 25 | 106.771 | | |
| Total | 51,107.00 | 30 | | | |

*Note:* Statistical significance levels: *p < .05, **p < .01, ***p < .001.
Abbreviations: SORDs, preschoolers with sensory over-responsivity disorder; SDQ, Strengths and Difficulties Questionnaire; Hyper.-Inatt., hyperactivity/inattention subscale; SRS, Social Responsiveness Scale; RBS-R, Repetitive Behavior Scale-Revised.

aimed at improving sensory integration may also support emotional and behavioral self-regulation.

Addressing emotional dysregulation, our study found that behavioral dysregulation was also associated with heightened emotional responses and diminished social interactions (Carpenter et al., 2019). Contrary to expectations, no differences were found in executive functions, specifically in inhibitory control, within the SORD group. This result may be due to the small sample size and the assessment methodology of executive functions based on parent-reported measures using the CHEXI, as opposed to behavioral task-based assessments of self-regulation, such as the HTKS-R.

We anticipated that the SORD group would exhibit significantly more intense sensory-seeking behaviors, such as repetitive actions, similar to those observed in individuals with ASD (Yuan et al., 2022); however, this was not observed. According to Dunn's model, which delineates atypical sensory processing and response, a category of "sensory seeking" is defined by a high neurological response threshold and a counteractive response strategy (Dunn, 1997; 2001). Individuals who fall into this category, particularly during childhood, often engage in compulsive behaviors to alleviate discomfort from ordinary sensory experiences, such as the sensation of wearing socks, common household odors and everyday sounds made by family members, and these behaviors can manifest independently, often preceding the emergence of intrusive thoughts or obsessions (Hazen et al., 2008; Tal et al., 2023). In contrast, the lack of observed differences in our study regarding repetitive behaviors, including compulsions, may stem from the small size of our sample, the exclusion of other psychiatric comorbidities or the possibility that sensory-seeking behaviors are associated with specific subtypes or clinical profiles and not with all sensory-processing deficits (Van Hulle et al., 2019; Yuan et al., 2022); they may not be associated with SOR, specifically defined as a low neurological threshold and a passive self-regulatory strategy in response to sensory stimuli (Dunn, 1997). Consequently, our sample may not have adequately represented these profiles, which could explain the absence of these expected behaviors.

Interestingly, despite excluding preschool children diagnosed with ADHD from our sample, significant differences in ADHD symptoms were observed between the groups. Moreover, ADHD symptoms were significantly correlated with both increased SOR severity and poorer self-regulation. This finding suggests that these symptoms may not merely represent subclinical ADHD characteristics but could instead be indicative of compensatory behaviors driven by SORD to balance heightened arousal levels. Behaviors such as poor impulse control, distractibility, difficulty focusing and inappropriate movement and touching may lead to challenges that resemble behavioral issues seen in ADHD (Keating et al., 2022). Therefore, caution is warranted when diagnosing ADHD in preschool-aged children, as these behavioral symptoms may develop secondarily to underlying disorders like SORD. Furthermore, children with early sensory processing and self-regulation impairments may be at greater risk of experiencing difficulties with perception, language and emotional or behavioral development during the preschool and school years (DeGangi et al., 2000). Long-term follow-up will be crucial in determining whether interventions aimed at improving self-regulation can also mitigate ADHD-like symptoms, advancing our understanding of the interconnectedness of these disorders and informing effective treatment strategies.

Despite the insights gained from this study, several limitations must be acknowledged. First, the relatively small sample size may constrain the generalizability of our findings and limit statistical power. Second, the reliance on parent-reported measures – particularly for assessing executive functions using the CHEXI – may introduce subjective bias and may not fully capture the complexity of cognitive regulatory processes. Third, the exclusion of children with psychiatric comorbidities, although essential for isolating the unique features of SORD, may have limited the representativeness of the sample, as many children with sensory processing challenges often present with overlapping psychiatric symptoms in clinical settings. Fourth, the cross-sectional design of the study precludes any causal inferences or temporal conclusions regarding the relationship between SOR and self-regulation difficulties.

Additionally, the study did not incorporate neurophysiological assessments, such as EEG (electroencephalogram), RSA or skin conductance, which could have provided objective data on sensory gating, autonomic regulation and arousal-related processes. While the preschool age group presents unique challenges in terms of compliance and data quality for such physiological measurements, the absence of these metrics limits the interpretation of the biological mechanisms underlying the observed behavioral patterns. Future research should prioritize the integration of multi-modal assessment approaches – including physiological, behavioral and neurocognitive data – within longitudinal frameworks to more comprehensively examine the developmental trajectory and neurobiological underpinnings of SORD.

In conclusion, this study underscores the distinct self-regulatory challenges experienced by preschool children with SORD, even in the absence of psychiatric comorbidities. Using a multi-method assessment approach – including the PAPA interview and observational evaluation with the CARS – we ensured diagnostic accuracy and the exclusion of ASD and other clinical conditions based on the DC:0–5 framework. Self-regulation was objectively measured through the HTKS-R task, which captures core executive components, such as cognitive flexibility, working memory and inhibitory control.

Our findings highlight the importance of recognizing SORD as a standalone clinical condition with distinct behavioral consequences, particularly in the domain of self-regulation. By focusing exclusively on the preschool period, this study contributes unique developmental insights into how early sensory processing difficulties manifest behaviorally. These results reinforce the need for early, targeted interventions that support regulatory skill development in young children with SORD – potentially reducing the risk of later academic, emotional and social difficulties. Future work should continue to refine diagnostic criteria and investigate the longitudinal outcomes of early self-regulation interventions in this underserved population.

**Open peer review.** To view the open peer review materials for this article, please visit http://doi.org/10.1017/gmh.2025.10076.

**Data availability statement.** The data set used can be accessed by contacting the corresponding author.

**Author contribution.** All authors declare that they have contributed to the conception or design of the work; the acquisition, analysis or interpretation of data; drafting or revising the manuscript and have given final approval of the version to be published, agreeing to be accountable for all aspects of the work.

**Competing interests.** The authors declare none.

**Ethical statement.** The research protocol was approved by the Human Research Ethics Committee of Ankara University School of Medicine (Approval number: İ05–383-2024), and the study was conducted in accordance with the Declaration of Helsinki.

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
