## [Reviewer Report]

This study aims to examine self-regulation difficulties in preschool-aged children diagnosed with Sensory Over-Responsivity (SOR). Although Sensory Over-Responsivity Disorder (SORD) is not currently included in most formal diagnostic systems, characterizing its clinical features and longitudinal outcomes in early childhood is crucial for advancing psychiatric nosology in this developmental period. In this regard, the manuscript addresses an underexplored yet clinically relevant domain and highlights an area of growing interest in early childhood mental health research.

While the manuscript is generally well-structured and the content is relevant, minor revisions are needed in terms of language and grammar to improve clarity and readability. A thorough proofreading by a native English speaker or a professional language editor is recommended to ensure consistency in academic tone and eliminate occasional grammatical inaccuracies.

The abstract is well-written, clear, and provides a concise summary of the study’s aims, methods, and main findings. No revisions are needed in this section.

In the introduction, the authors present a clear rationale regarding the challenges children with SORD may experience, including emotional and behavioral difficulties. However, the first hypothesis posits that no significant differences will be observed between children with and without SORD in terms of emotional and behavioral problems. This seems somewhat inconsistent with the literature presented earlier. The authors are encouraged to clarify the theoretical or empirical basis for this hypothesis.

Hypothesis 3 suggests that children with SORD are expected to exhibit increased sensory-seeking behaviors, including repetitive or compulsive behaviors. However, this assumption may require further theoretical justification. According to the existing literature, sensory over-responsivity is typically associated with sensory-avoiding or sensory-sensitive behaviors rather than sensory-seeking. If the authors are conceptualizing a distinct behavioral phenotype or referencing prior findings that support this association, it would be helpful to elaborate on this point in the introduction or discussion. While the authors reference Dunn’s sensory processing framework in the discussion, it would strengthen the manuscript to more explicitly situate SORD within this model—particularly by emphasizing its association with the sensory avoiding or sensory sensitive profiles.

The manuscript would benefit from a more detailed description of the control group recruitment process. Specifically, it would be helpful to indicate from which clinic or population the control group was selected, and what inclusion/exclusion criteria were applied.

While the current findings are promising, expanding the sample in future studies would enhance statistical power and generalizability. Moreover, longitudinal follow-up examining the potential emergence of psychiatric conditions in children with SORD would provide valuable insight into the developmental trajectory and clinical implications of early sensory over-responsivity. If this study is part of a broader ongoing project, it would be helpful to briefly indicate that in the conclusion. Such a note would clarify the potential developmental scope of the research and strengthen its contribution to the field.

Aside from the issues noted above, the discussion is well-developed. Addressing such a rarely studied but clinically important topic in the preschool period is commendable and has the potential to enrich the literature. The manuscript is suitable for minor revision and re-evaluation.

---

## [Reviewer Report]

The research is interesting and new as it adresses a problem that clinicans in child psychiatry field face every day.The research is about an overlooked subgroup of children who dont have psychiatric diagnosis but have SORD. Their effort to identify the differences in this group may help these children to prevent further pscychiatric problems or misdiagnosis. Also self regulation and sensory seeking behaviour is another different area that needs to be addressed. Valid Turkish version of a structured evaluation for prechoolers is also a strenght of the study as the diagnosis part is the most important step in the study. Especially the methodology part is well designed for their research purposes. However, I have some suggestions to make the paper better. First, the introduction part may have some neurobiological insight for establishing the relationship between self regulation and SORD. If it is their conclusion that they are related, then it may be added, too. Second,discussion lacks the contrary findings in the literature. Only their hypothesis and their possible explanations for negative results are discussed as their limitations. For example; if there are some contrary views, study results for describing SORD as another subgroup, they should be mentioned and the reason for not they are agreeing with them depending on their results should be added. I think the paper would be better to be published after these arrangements.

I can review the paper again after the response of the authors.

Warm rigards

---

## [Editor Report]

Dear Uygun,

Your manuscript: “Exploring Self-Regulation Deficits in Sensory Over-Responsivity Disorder: A Preschool Comparative Analysis” has now been reviewed,

---

## [Reviewer Report]

The manuscript is considered suitable for publication, provided that the recommended revisions have been adequately addressed.

---

## [Editor Report]

Dear Uygun,

Your revised manuscript :’Exploring Self-Regulation Deficits in Sensory Over-Responsivity Disorder: A Preschool Comparative Analysis', has now been reviewed